# Melanin—The Éminence Grise of Melanoma and Parkinson’s Disease Development

**DOI:** 10.3390/cancers15235541

**Published:** 2023-11-23

**Authors:** Danuta Krasowska, Agata Małek, Joanna Kurzepa, Lucyna Kapka-Skrzypczak, Dorota Krasowska, Jacek Kurzepa

**Affiliations:** 1Department of Medical Chemistry, Medical University of Lublin, 20-059 Lublin, Poland; agata.malek@student.umlub.pl (A.M.); jacek.kurzepa@umlub.pl (J.K.); 21st Department of Medical Radiology, Medical University of Lublin, 20-954 Lublin, Poland; joanna.kurzepa@umlub.pl; 3Department of Molecular Biology and Translational Research, Institute of Rural Health, 20-090 Lublin, Poland; kapka.lucyna@imw.lublin.pl; 4World Institute for Family Health, Calisia University, 62-800 Kalisz, Poland; 5Department of Dermatology, Venereology and Pediatric Dermatology, Medical University of Lublin, 20-059 Lublin, Poland; dorota.krasowska@umlub.pl

**Keywords:** eumelanin, pheomelanin, neuromelanin, melanoma, Parkinson’s disease, dopamine

## Abstract

**Simple Summary:**

This article examines the fascinating association between melanoma, a malignant skin cancer, and Parkinson’s disease (PD), a neurodegenerative disorder. Both diseases involve cells that produce melanin, a pigment that provides skin color and protects against UV radiation. This study explores the potential impact of melanin synthesis on these diseases, considering the divergent roles of eumelanin and pheomelanin, melanin types present in both skin and brain cells. Additionally, it investigates the influence of PD treatments, such as L-DOPA, on melanoma risk, although the nature of this relationship remains uncertain. The research aims to provide insights into these intricate connections and their implications for the medical field.

**Abstract:**

A common feature of Parkinson’s disease (PD) and melanoma is their starting points being based on cells capable of converting tyrosine into melanin. Melanocytes produce two types of melanin: eumelanin and pheomelanin. These dyes are designed to protect epidermal cells from the harmful effects of UV radiation. Neurones of the substantia nigra, which degenerate during PD, produce neuromelanin, the physiological role of which is not fully explained. This article discusses the potential role of melanins in the pathogenesis of both diseases. Melanins, due to their ability to accumulate toxic substances, may become their sources over time. The use of glutathione for the synthesis of pheomelanins and neuromelanins may reduce the antioxidant capacity of cells, leading to an excessive synthesis of free radicals. This study also tested the hypothesis that certain drugs used in the treatment of PD (L-DOPA, MAO-B and COMT inhibitors, and amantadine), aimed at increasing dopamine concentration, could potentially contribute to the development of melanoma. The role and properties of melanins should continue to be researched. Whether excessive melanin synthesis or its accumulation in the extracellular space may be factors initiating the development of diseases remains an open question.

## 1. Introduction

Melanoma is one of the most serious human cancers. It develops as a result of the neoplastic transformation of melanocytes, cells that are a part of the basal layer of the epidermis, whose main physiological role is the production of skin pigments—melanin [1]. Over the past 60 years, the incidence rate of melanoma has been gradually increasing [2]. The incidence varies from 5 to 7 cases per 100,000 inhabitants per year in Italy, through to about 20 cases per 100,000 inhabitants in the USA, to the highest values in other countries such as over 50 cases per 100,000 inhabitants in Queensland (Australia) [3]. The most important modifiable risk factors for melanoma include exposure to ultraviolet radiation (the role of radiation in the pathogenesis of melanoma is discussed below). However, the etiology of melanoma is multifactorial and results from the interaction between genetic susceptibility and environmental exposure. Fair skin, blue or green eyes, and freckles are markers of greater susceptibility to melanoma [4].

Parkinson’s disease (PD) is a neurodegenerative disease in which the leading pathological feature is the degeneration and loss of nerve cells. The incidence of PD ranges from 1 to 2 per 1000 in unselected populations, increasing with age and therefore affecting 1% of the population over 60 years of age. It is rare before the age of 50 and occurs in approximately 4% of the oldest people [5]. During idiopathic PD, the dopaminergic neurons located inside the substantia nigra of the midbrain undergo atrophy [6].

What connects both types of cells, melanocytes and neurons of the substantia nigra, is the ability to gradually convert tyrosine into different kind of melanin. Before melanin is formed, the addition of a second hydroxyl group to the tyrosine aromatic ring produces the DOPA compound, which can be decarboxylated into the first of the series of catecholamines—dopamine (DA). In other cells (e.g., in the adrenal medulla), dopamine is then hydroxylated to noradrenaline or further methylated to adrenaline. The oxidation of both hydroxyl groups in DOPA converts it to a very reactive DOPAquinone, which, in further transformations, polymerizes to give just two types of skin melanin [7]. Eumelanin does not form a homogeneous compound; it is a mixture of nitrogenous polymers of 5,6-dihydroxyindole (DHI) and 5,6-dihydroxyindole-2-carboxylic acid (DHICA) units [8].

The disappearance of dopamine results in the clinical symptoms observed in PD—muscle stiffness, tremors, and the disturbance of precise movements. In this study, we wanted to consider whether tyrosine metabolites (melanin) have a specific role in the pathogenesis of both diseases. Moreover, it is not entirely clear whether PD treatment associated with increasing dopamine levels may influence the development of melanoma.

## 2. Origin of Melanocytes and Neurons of the Substantia Nigra

One of the common features of melanocytes and neurons, including dopaminergic neurons, is their ectodermal origin [9]. The melanocyte lineage originates from the neural crest cells. These cells are highly migratory, forming many specialized structures and tissues in the developing embryo through migration, proliferation, and differentiation [10]. The research of Erickson and Goins using avian transplants has shown that cells that leave the crest early follow a ventral migration path through the anterior sclerotome and become neurons and glia, whereas late cells follow a dorsolateral route between the ectoderm and somites through the developing dermis and they become melanocytes. In addition, the study showed that the fate of cells is not determined by their migration path, but it is determined before leaving the crest [11]. The migration through the developing embryo and interaction with their environment containing various types of cells to finally become the stem cell population through which they self-renew might be one of the reasons that melanoma is particularly aggressive and metastatic [10].

During PD, the degeneration of dopaminergic neurons is most severe in the ventral part of the substantia nigra pars compacta, compared with other subregions of the substantia nigra that seem to be more resistant to the neurodegeneration process [12]. A study by Marchand and Poirier published 40 years ago in *Neuroscience* indicates that neurons in the substantia nigra in rats arise at two different points on the basal plate at the level of the foveal isthmus (the meso–isthmus junction) and migrate radially as two separate streams towards the ventral midbrain during embryogenesis [13]. Using tyrosine hydroxylase, the main enzyme for catecholamines synthesis, as a marker, Aubert et al. described the migration route of dopamine-synthesizing cells during human embryogenesis. They showed that cells capable of synthesizing dopamine appear for the first time in the midbrain in the 12th week of fetal life, migrating to the target site to reach the substantia nigra location after approximately 7 weeks [14]. However, not only the place of origin, but also various transcription factors, which translate into different protein expressions, may have decisive impacts on the susceptibility to neurodegeneration of mature neurons (this aspect was clearly described by Fu et al. [12]).

## 3. Eumelanin, Pheomelanin and Neuromelanin; Do Dyes Cause Trouble in the Cell?

Melanocytes synthesize two types of melanin, brown–black eumelanin, and red pheomelanin [15]. Neuromelanin is produced in neurons of the substantia nigra. It is a condensation product of two units present in both eumelanin and pheomelanin [16] Figure 1.

Skin melanin is synthesized in melanosomes, and then it is transported to neighboring cells, keratinocytes, whose resources are located in supranuclear “caps” that protect against harmful ultraviolet radiation (UV). Several review publications discuss the role and distribution of melanin in different human races, so we will not duplicate this information which is provided in the relevant articles [18]. It is undoubtedly a fact that UV radiation is the main risk factor for the development of melanoma [19]. UV radiation may cause DNA damage and lead to the formation of various types of mutagenic compounds. Direct DNA damage is caused by UVB (~280–320 nm of wavelengths) more than UVA (~320–400 nm of wavelengths). The second one seems to be less dangerous and carries less energy than UVB but penetrates deeper layers of the skin. The mechanism of the harmful effect of UVA is the generation of free oxygen radicals rather than the direct degradation of DNA (for a review, see ref. [20]). However, the action of UV rays leading to the formation of melanoma is undoubtedly more complicated than the stimulating of ROS and generating of mutagenic compounds. UV is involved in the activation of various cytokines (for example, interferon gamma) and pro-inflammatory enzymes (for example, matrix metalloproteinase-9) [19]. During increased exposure to UV radiation, there is an intense synthesis of melanin in the skin leading to better protection against radiation. The protective effect of melanin is confirmed both in experimental and epidemiological studies. Previous research shows that melanin successfully protects against electromagnetic radiation, causing the dispersion of optical energy and converting it into heat [21]. In addition, it has been shown that melanins are able to absorb other, more energetic types of electromagnetic radiation, such as X-rays and Gamma rays. Some species of melanin-producing fungi colonize environments with a high degree of radiation (e.g., nuclear reactors) without significant influence on their viability [22]. One of the mechanisms of ionizing radiation, especially electromagnetic radiation, is water radiolysis, which results in the formation of a number of free radicals from the breakdown of water molecules [23]. However, melanins that have the ability to scavenge free radicals may play a potential role in protection even against ionizing radiation-causing water radiolysis. The protective effect of melanin has also been confirmed in human observations. People with dark skin have a several dozen times lower risk of melanoma [24].

Since melanin acts as a protective shield against UV rays and has a free radical scavenging effect, why does melanoma arise mainly in cells where the concentration of this pigment is high? In addition to genetic factors, we want to consider what role melanin may play in the induction of carcinogenesis. Cells laden with melanin should theoretically have a lower tendency to form cancer. Perhaps the key here is the imbalance between synthesis of different types of melanin, eumelanin, and pheomelanin. Pheomelanin requires the sulfhydryl group, of which an important source in the body is glutathione, a compound with significant antioxidant properties. Therefore, the intense pheomelanin synthesis can lead to a depletion of glutathione in cells, potentially exposing cells to free radical attack. Research by Tanaka et al. showed that the benzothiazole moieties of pheomelanin showed pro-oxidative effects when exposed to UVA. The level of reduced glutathione (GSH) was also decreased [25]. Lembo et al. point out that there is growing evidence that melanin, particularly pheomelanin, is involved in the development of melanoma and that this process does not require UV exposure for initiation. In their experiment, the red hair pheomelanin and, in a lesser degree, the black hair eumelanin, but not the white hair protein, significantly increased pro-inflammatory cytokines and decreased cell viability in the in vitro model [26]. Interestingly, the research of Premi et al. published in *Science* indicates that the mutagenic effect of melanin can occur under the influence of free radicals, without the necessary participation of UV [27]. In addition, due to the specificity of oxidoreductive processes, eumelanin may also exhibit oxidative properties, which significantly complicates the unclear role of both pigments in the formation of melanoma [28]. It is also worth remembering that melanins have a high affinity for heavy metals, the presence of which may have a negative impact on cell metabolism and intensify oxidative stress [29,30]. However, binding heavy metals can have the opposite effect on cells; it is harmful in the long term, but it is a safe way to remove heavy metals from cells [31]. The proposed role of melanin in the development of melanoma is presented in the Figure 2.

The role of neuromelanin in the pathogenesis of PD is also unclear. Since idiopathic PD only affects the neurodegeneration of neuromelanin-producing cells, the relationship of this pigment to the disease has been the subject of research for many years [32]. The main role of tyrosine in the neurons of the substantia nigra is to be a substrate for the conversion to dopamine. DA is a key neurotransmitter of the extrapyramidal system, the deficiency of which is responsible for the symptoms of PD [33]. The substantia nigra gives connections to the striatum and is part of the extrapyramidal system responsible, among others, for motor skills and control of voluntary movements [6]. Together with DA, neuromelanin is synthesized in neurons, giving a black color to the cells (the name comes from the black color of the cells caused by neuromelanin). Neuromelanin is also present in noradrenergic neurons, but surprisingly it does not occur in cells synthesizing the last of the series of catecholamines—adrenaline [34]. It seems that neuromelanin is a byproduct of dopamine metabolism formed in the auto-oxidation pathway, without the key role of UV radiation in this process (opposite to skin melanin) due to the obvious limitation of UV access to neurons in the midbrain [35]. The enzyme that oxidizes tyrosine to L-DOPA (as tyrosinase) is not necessary for the synthesis of neuromelanin, because neuromelanin is also present in albinos with a genetic defect of tyrosinase [36]. In addition, tyrosinase activity is negligible in the normal human brain. However, rats stereotactically implanted with a human tyrosinase viral vector into the substantia nigra significantly increased neuromelanin synthesis, suggesting a possible, but not necessary, role of this enzyme in neuromenaninogenesis [37].

An interesting observation is the presence of neuromelanin mainly in humans, to a lesser extent in primates, and almost completely absent in other species. Since many non-neuromelanin-producing animal species have numerous catecholamine-synthesizing neurons, it follows that the conversion of tyrosine to dopamine or noradrenaline in the cell need not be correlated with simultaneous neuromelanin synthesis [34].

Neuromelanin concentration fluctuates in the aging brain. It seems that exceeding a certain concentration threshold of this dye in dopaminergic neurones may trigger the neurodegeneration. The animals mentioned above developed movement disorders characteristic of PD only after synthesizing a certain amount of neuromelanin [37]. The neuromelanin level increases with the age of people from a negligible amount in the first year of life to over 3 ug per 1 mg protein of substantia nigra pars compacta [38]. However, the mere presence of neuromelanin is not the only reason for the development of PD symptoms, because as symptoms increase, the amount of pigment is reduced, which has been confirmed in pathological [38] and radiological studies using new neuromelanin imaging techniques in magnetic resonance imaging (MRI) [39]. Neuromelanin present in cells seems to have a protective function against toxins entering dopaminergic neurons, binding them and preventing their toxicity [40]. On the other hand, the binding of toxins within neuromelanin causes the concentration of toxins inside cells containing this pigment to reach high values. D’Amato’s 1987 study indicates that the neurotoxin MPTP (1-methyl-4-phenyl-1,2,3,6-tetrahydropyridine), after entering the body, binds inside the dopaminergic neurons to neuromelanin, which becomes an intracellular depot of this toxin, gradually releasing it. At the same time, the loading of neuromelanin with chloroquine, thereby reducing the affinity of the dye for toxins, reduces the intracellular MPTP reservoir, reducing extrapyramidal symptoms in experimental animals [41]. Also of interest is the increase in the amount of extracellular neuromelanin in the elderly, which has been demonstrated in histopathological studies. The toxins and metal ions accumulated in neuromelanin can be released, increasing oxidative stress in the extracellular space that could potentially trigger or intensify cell degeneration [42]. The possible stages and mechanisms of development of dopaminergic neuron degeneration are presented in Figure 3. However, the pathogenesis of PD is probably not based solely on neuromelanin metabolism. Symptoms similar to PD can be induced in animals, which, as mentioned, have a much lower amount of neuromelanin than humans. Moreover, animal models of PD are based either on modifications of PD-related genes or on the administration of toxins that selectively attack substantia nigra cells without directly affecting the neuromelanin metabolism [43].

As shown in Figure 1, neuromelanin synthesis requires glutathione, a tripeptide with strong antioxidant properties. In the 1990s, post-mortem studies showed reduced concentrations of glutathione in neurons of the substantia nigra pars compacta of patients with PD [44]. In addition, it was found that in patients with PD, the concentration of glutathione in other brain regions did not change significantly [45]. This indicates that glutathione is involved in the pathogenesis of PD. However, it is not known whether glutathione depletion is the cause of the disease or its effect. The reason for the decrease in glutathione concentration is also unknown. It can be assumed that the use of glutathione for the synthesis of neuromelanin, together with increased oxidative stress caused by the release of substances accumulated in the dye, may have a role in initiating the neurodegeneration of dopaminergic cells. On the other hand, an attempt to explain the pathogenesis of melanoma and PD with a reduced amount of glutathione seems too simple and does not explain why only some cells of the body would react with pathological changes to its lack.

An interesting issue is the role of vitamin D in both the pathogenesis of melanoma and PD. Vitamin D is a known steroid compound, the deficiency of which may adversely affect various neurological diseases such as stroke, multiple sclerosis, Alzheimer’s disease and PD [46]. In the case of PD, it was noticed that patients tend to have a lower concentration of vitamin D in serum and bone, changes indicating that, for PD patients, a deficiency of this vitamin appears more often. Previous clinical studies have shown that vitamin D supplementation in PD patients not only improves bone condition, but also has a positive effect on reducing neurological symptoms typical of PD [47]. The anticancer effects of vitamin D have been documented in various cancers in both pre-clinical and clinical studies. There are reports in the literature that a higher concentration of vitamin D in serum is associated with a better prognosis in patients with melanoma [48]. However, the relationship between vitamin D and melanoma seems to be more complex, including the role of UV radiation, which is necessary for the synthesis of vitamin D and, at the same time, is a risk factor of melanoma. Melanin, which is a natural sun filter, effectively absorbs UV radiation, significantly limiting the skin photosynthesis of vitamin D3. The dark melanin pigment of Africans reduces the skin’s ability to produce the previtamin D3 by 95–99% compared to Caucasians [49].

## 4. Treatment of PD and Development of Melanoma

Epidemiological studies show that, in general, both patients with PD are less likely to develop various types of cancer and patients with cancer are less likely to develop PD [50]. This result seems logical, because cancer is a proliferative disease caused by excessive cell division, while PD, as a degenerative disease (also containing melanin), is dominated by cell loss. However, this rule does not apply to all cancers. Previous observations indicate a certain positive connection between PD and melanoma and nonmelanoma skin cancers. According to data from the Rochester Epidemiology Project, PD patients were 3.8 times more likely to have a preexisting melanoma compared with controls (95% CI, 2.1–6.8; *p* < 0.001), and the risk of developing PD was 4.2 times higher in patients with melanomas (95% CI, 2.0–8.8; *p* < 0.001) [51]. A study based on data from the Danish National Hospital Register showed that melanomas in patients with PD arose 95% more often than could be expected (44 observations, compared with 22.5 cases that could be expected) [52]. A meta-analysis covering over 140,000 individuals showed a higher incidence of skin cancer in patients with PD (OD = 1.25, 95% CI: 1.17–1.33; *p* < 0.0001) [53]. The observed relationships can be explained in many ways. One of them is a genetic predisposition to an increased risk of both types of disease occurring simultaneously. An exemplary trigger factor is a protein, alpha-synuclein, which is associated with the pathogenesis of dopaminergic cell neurodegeneration in PD and is also involved in melanomagenesis [54]. Certainly, common biochemical processes present in both dopaminergic neurones and melanocytes include the transformation of tyrosine to melanin formation. Therefore, theoretically, the treatment used in PD, which increases DA concentration, could contribute to the development of melanoma in some way. However, it is not known whether the increase in the incidence of melanoma in PD patients is related to the increase in melanin synthesis in the skin.

The cause initiating the degenerative process of substantia nigra cells of the midbrain is not known. The fact is that it can be observed—a gradual loss of dopaminergic neurones, and thus a decrease in DA concentration mainly in the striatum, to which the axons of the substantia nigra reach [6]. Therefore, a strategy for the treatment of PD is to increase the concentration of DA in the brain. This type of therapy directly improves the clinical condition of patients by reducing the clinical symptoms of the disease. In this paper, we wanted to focus on the selected substances used in PD, the action of which aims at increasing the concentration of DA in the brain. The first is L-DOPA, a direct precursor of DA. The second substance, amantadine, elevates DA concentration by increasing the release of DA from neurones [55]. The last two agents inhibit the breakdown of DA via monoamine oxidase type B, MAOB (rasagiline) or via catechol-O-methyltransferase, COMT (capone family, e.g., entacapone) [56,57]. All of them finally increase the DA level within the central nervous system, acting through different mechanisms (see Figure 4).

To answer the question of whether PD treatment intensifies melanogenesis, leading in some cases to the development of melanoma, relevant publications were searched in the PubMed database by entering the phrase “melanoma” and “L-DOPA” (or “levodopa”, “amantadine”, “MAO”, “COMT”), respectively, in the title or the abstract. In the case of MAO or COMT inhibitors, the specific drug names were omitted to increase the number of publications found. The search covered the last 20 years.

## 5. L-DOPA

Among the above-mentioned pharmaceutical agents, L-DOPA has the greatest direct connection with the melanin synthesis pathway, which may potentially be associated with the pathogenesis of melanoma. L-DOPA is a direct precursor of DA and can be converted to other compounds involved in melanin synthesis. Furthermore, in vitro studies indicate that L-DOPA (similarly to tyrosine) is not only a substrate for melanogenesis, but it is a positive regulator of several processes that occur inside the cell (for a particular review see ref. [58]). The rate of a tyrosinase-catalyzed reaction increases in the presence of L-DOPA [59]. At the same time, an increase in tyrosinase activity was found in melanoma [60]. It is also worth emphasizing that some publications from the last three decades of the 20th century revealed a possible co-occurrence of L-DOPA treatment and the risk of melanoma [61]. This may indicate a direct relationship between the availability of the tyrosinase substrate (L-DOPA) and the development of this kind of tumor. However, the relationship between PD treatment with exogenous L-DOPA and the intensification of melanogenesis or the risk of melanoma is unclear. Searching for the words “L-DOPA” (or “levodopa”) and “melanoma” in the title or abstract over the last 20 years reveals 103 publications. To increase the number of retrieved publications, the search was not limited only to L-DOPA used as a therapeutic agent in PD, as all publications regarding L-DOPA in melanoma were searched. Most of the studies found did not address the effect of L-DOPA treatment on the development of melanoma but numerous studies were mainly related to the in vitro testing of tyrosinase, in which L-DOPA was used as a substrate for this enzyme. However, after screening the received publications, only six of them were included in the analysis. The first one presented a case report of a patient treated with L-DOPA (together with carbiDOPA), who developed eruptive melanocytic nevi (EMN) associated with the rapid development of multiple melanocytic nevi on the skin. The authors of the case hypothesized that the relative increase in L-DOPA during the treatment of PD may stimulate the development of melanocytic nevi production, leading to the development of EMN [62]. EMN is not a melanoma, but the presented case report indicates the possibility of proliferation of melanin-containing cells under the influence of L-DOPA. In a letter to the editor published in 2021, a clinical case of a patient treated with L-DOPA due to PD, who developed melanocytic hyperactivation simulating acral lentiginous melanoma, was described. Due to clinical and histopathologic discordance, the whole skin change was removed [63]. Bougea et al. reported in a meta-analysis that melanoma is the most common skin disease associated with the use of L-DOPA. This conclusion was made on the basis of a review of clinical cases in which of 32 described patients, 13 were diagnosed with melanoma after taking L-DOPA [64]. However, Olsen et al. showed that L-DOPA had no effect on the risk of malignant melanoma, as indicated by an odds ratio of 1.0 (95% confidence interval, 0.8–1.3) per 1000 g of cumulative drug intake. The authors concluded that the increased incidence of malignant melanoma observed in PD patients was limited to the patients with idiopathic PD, but was not related to L-DOPA treatment [65]. Another study describing several families with a genetic predisposition for developing melanoma due to the germline mutation showed that they may had an increased risk of developing this cancer after starting L-DOPA therapy. The authors suggested that there is a need to reconsider the hypothesis of the involvement of L-DOPA in the development of melanoma, at least in the context of the high-risk genetic basis [66]. In research published by Constantinescu et al., the incidence of malignant melanoma was found to be higher than expected in PD patients. However, there was no association between L-DOPA treatment and the occurrence of melanoma [65]. In a letter to the editors of the Lancet, Zanetti and Rosso concluded the previous reports on the relationship between L-DOPA treatment and the occurrence of melanoma, stating that there is no direct relationship. The aetiology of both diseases (PD and melanoma) is multifactorial, and despite the demonstrated relationship between the occurrence of both diseases, no such relationship between PD treatment and the occurrence of melanoma can be demonstrated [67].

Observations so far do not allow us to conclude that there is a clear connection between L-DOPA treatment and excessive melanin synthesis or the development of melanoma. Providing an exogenous substrate for tyrosinase is not sufficient to notice an obvious increase in melanin synthesis, and individual reports of hyperpigmentation are insufficient to confirm such a relationship. From a biochemical point of view, this is understandable and observed in relation to other enzymes. For example, the in vivo supply of arginine, a substrate for the synthesis of nitric oxide, did not increase the production of nitric oxide and citrulline, which are the products of this reaction [68]. However, melanocytes have the ability to capture extracellular L-DOPA and convert it into melanin [69], so a potential risk can be assumed that large doses of this drug will increase the synthesis of melanin in skin cells.

## 6. Amantadine, MAO-B and COMT Inhibitors

Amantadine is a drug commonly prescribed to mitigate the motor symptoms associated with PD, primarily through its action as a receptor antagonist of N-methyl-D-aspartate (NMDA). These effects, particularly in relation to melanin production, are of interest as dopamine regulates melanin, the pigment responsible for skin, hair, and eye color. Amantadine affects dopamine levels in multiple ways. It enhances the release of dopamine by blocking the N-methyl-D-aspartate (NMDA) receptor and inhibits the reuptake of dopamine, thereby increasing its availability in the synaptic cleft. Given these dopaminergic effects, it is tempting to speculate that amantadine could influence melanin production [70].

When searching for publications published in the last 20 years on amantadine and melanoma, we found seven items. None of them concern clinical trials. After rejecting the review articles (three publications) and the article in Japanese, there were three items left, only one of which directly concerned amantadine and melanoma. The publication concerned in vitro studies related to the anti-cancer effect of amantadine on melanoma cells. The study had shown that amantadine enhances the effect of mitoxantrone and cisplatin by inducing apoptosis in the melanoma cell line. However, there is no direct reference in the work to the effect of amantadine on melanin synthesis in melanoma cells [71].

When searching for publications on rasagiline and melanoma in the PubMed database, four articles appeared; two original works, one review and one letter to the editor. Only the first two related to the relationship between rasagiline and melanoma. In a retrospective study, Johannes et al. showed that patients treated with rasagiline had a slightly higher number of melanomas. However, the authors believed that this was due to increased dermatological supervision in these patients, compared to patients treated with other preparations [72]. The second study compared the effect of rasagiline administered orally versus transdermally in mice implanted with human melanoma cells. The aim of the study was to determine whether the direct effect of an MAO-B inhibitor on the skin would contribute to the development of melanoma and how both types of treatment would affect the tumor mass. Both types of treatment reduced the tumor mass, which led the authors to conclude that rasagiline may be a candidate for an antimelanoma drug [73].

Searching for the words “COMT” and “melanoma” in the title or abstract for the last 20 years reveals five publications. The assumed search criteria for publications indicate that the topic of the relationship between treatment with COMT inhibitors for PD and the risk of melanoma has not been widely discussed in the literature. One of the retrieved publications concerned in vitro studies in which the effect of UVB on COMT activity in melanoma cells was assessed. The study showed the presence of COMT in both melanoma cells and keratinocytes. Under the influence of UVB, there was a significant reduction in the activity of this enzyme in melanoma cells. Furthermore, the addition of the COMT inhibitor, tolcapone, reduced melanin levels in melanoma cells parallel to reduced cell numbers [74]. The observations may seem unexpected, as COMT inhibition potentially should increase the availability of DA, which further can be used for melanin synthesis. However, this study shows that under in vitro conditions, COMT activity is an important factor that stimulates melanin synthesis in melanoma cells. The result of the study can be compared with the study of the rasagiline effect on the growth of melanomas, which also surprisingly showed that instead of the expected increase in tumor cells, the use of an MAO-B inhibitor reduced their number [73].

## 7. Future and Conclusions

The unique ability of both melanocytes and dopaminergic neurons of the substantia nigra to synthesize melanin allows us to assume that pathological processes affecting both types of cells may have a common factor, which is melanin. However, the assumption that an excessive amount of pigment in various types of cells leads to degeneration (PD) or pathological proliferation (melanoma) seems too bold. Future in vitro studies that attempt to elucidate the role of melanin in the pathogenesis of PD may involve assessing the effect of neuromelanin on the survival of various types of cells in the nervous system. Such a study may answer the question of whether the process of neurodegeneration under the influence of neuromelanin is typical for cells in the substantia nigra or whether other cells may be subject to it. On the other hand, the protective role of melanin against UV radiation may disturb the apoptosis of cells which, having a limited ability to die, over-proliferate and give rise to melanoma.

## Figures and Tables

**Figure 1 cancers-15-05541-f001:**
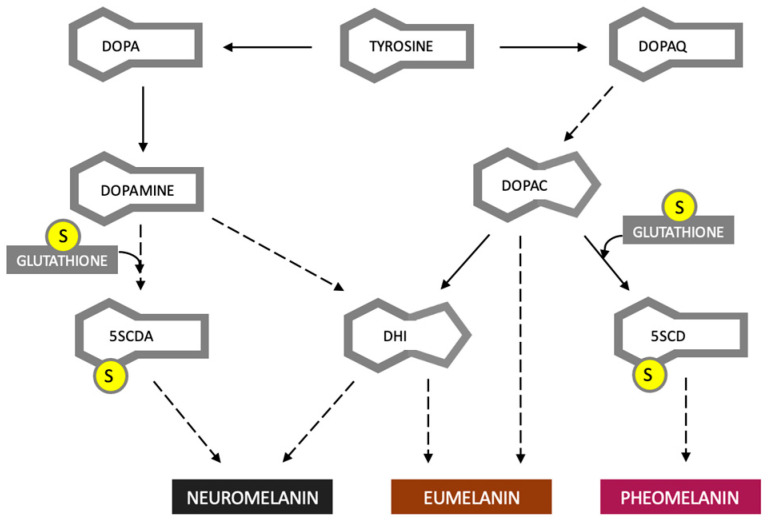
Simplified scheme of melanin synthesis. Tyrosine is a precursor to DOPA and Dopaquinone (DOPAQ). In dopamine-producing cells, there is a partially spontaneous conversion of dopamine to 5,6-dihydroxyindole (DHI), which can also be formed directly from Dopachrome (DOPAC). The synthesis of neuromelanin and pheomelanin requires glutathione. (5SCD) 5-S-cysteinyldopa, (5SCDA)5-S-cysteinyldopamine. Dashed line arrows represent several reactions. The letter “S” in a circle represents the sulfur of the thiol group. Based on [16,17].

**Figure 2 cancers-15-05541-f002:**
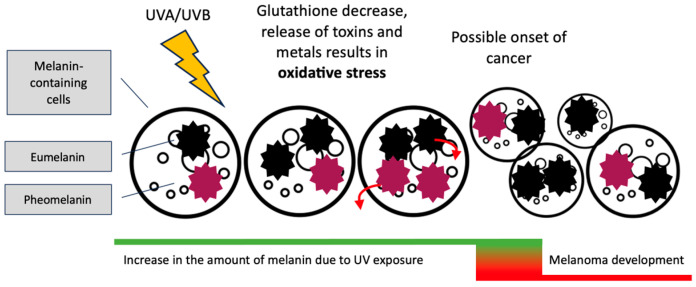
Proposed mechanism for the role of eumelanin and pheomelanin in the development of melanoma. The amount of melanin increases under the influence of UV radiation. Melanins have protective properties against UV rays, but the reduction of the glutathione level used for the synthesis of pheomelanin can induce oxidative stress [25]. In addition, melanins accumulate metal ions and toxins, which, when released gradually (red arrows), can potentially increase oxidative stress [29]. Certainly, this is not the only factor determining the development of melanoma.

**Figure 3 cancers-15-05541-f003:**
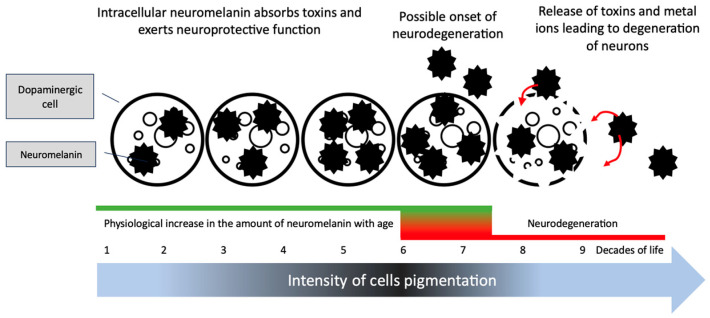
Proposed mechanism of neurodegeneration of neuromelanin-producing cells. During life, the amount of intracellular and extracellular neuromelanin increases, protecting cells from damage. After a certain threshold, accumulated toxins and metal atoms can be released (red arrows), leading to increased oxidative stress and cell degeneration. The mechanism that triggers the cascade of biochemical processes leading to neurodegeneration is unknown. Based on [38,42].

**Figure 4 cancers-15-05541-f004:**
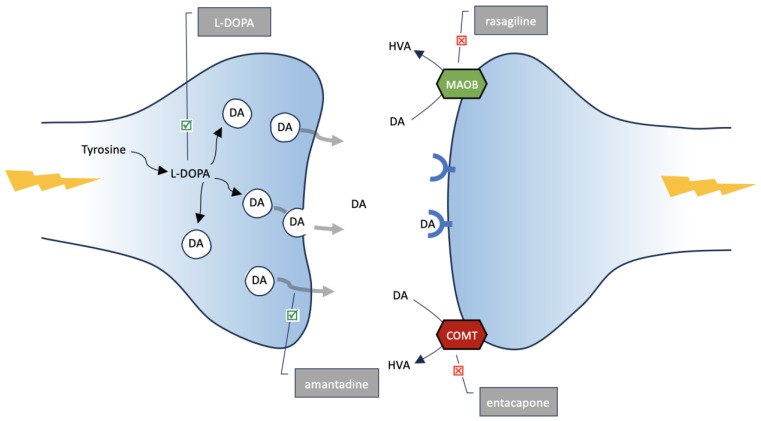
Scheme of action of L-DOPA, amantadine, rasagiline and entacapone leading to an increase in DA concentration in the synaptic space. In the presynaptic neurone, tyrosine is converted to L-DOPA and then to DA, which is placed in the synaptic vesicles waiting for a signal to be released outside the cell. In the synaptic space, DA binds to receptors in the membrane of the postsynaptic neurone, transmitting the impulse. L-DOPA is a direct precursor of DA, increasing its concentration in the brain. Amantadine increases DA secretion outside the cell. Rasagiline inhibits the breakdown of DA to homovanillic acid (HVA) by monoamine oxidase type B (MAOB), and entacapone inhibits the breakdown of DA by catechol-O-methyltransferase (COMT).

## Data Availability

The data are available upon request.

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
