# Peer review of "Melanin—The Éminence Grise of Melanoma and Parkinson’s Disease Development"

_cancers, 2023, doi:10.3390/cancers15235541_

Round 1

Reviewer 1 Report

Comments and Suggestions for Authors

Dear Authors!

I found your manuscript really well-written and organized. 

Just a minor point:

I suggest write 1-2 more paragraph in the introduction section referring with detail in the characteristics (epidemiology, frequency, cell abnormalities etc.) of melanoma and Parkinson's disease. 

Comments on the Quality of English Language

Really well-written. Good quality of English.

Author Response

I found your manuscript really well-written and organized. 

Just a minor point:

I suggest write 1-2 more paragraph in the introduction section referring with detail in the characteristics (epidemiology, frequency, cell abnormalities etc.) of melanoma and Parkinson's disease. 

Thank you very much for your positive assessment of our work.

Relevant paragraph on characteristics (epidemiology, frequency). We decided not to describe the pathological changes in melanoma cells and neurones affected by neurodegeneration, because in our opinion this description would deviate too much from the main purpose of the work.

Reviewer 2 Report

Comments and Suggestions for Authors

Major comments

The authors reviewed the literature on melanin and neuromelanin. The melanin pathway is not well described and need a better description. There is no method section with a clear description of which years were considered with which criteria, instead of writing them in the text. The medical report of the literature on the effect of PD treatment on the melanin pathways and melanoma should be more rigourous, with exactly for each treatment, how many paper were considered, how many reports with positive and negative results, this could be reported in a table with the number of patients, the reference included.

Then a discussion of the results can be included.

Minor comments

L41: the authors are general but talk about neuromelanin.

L47: « just two types of melanin” it is not totally correct because eumelanin is a mixture of nitrogenous polymers of DHI and DHICA, for reference see e.g.: ( Ito and Wakamatsu 2008). Melanin pathway could be better explained.

L47: the sentence on the PD symptoms could start a new paragraph.

L101: Pheomelanin requires sulfhydryl group of any origin, e.g.: glutathione or cysteine.

L108: reference is missing.

L108: melanin is not a protein.

L111: the authors could be cited instead of the publication journal.

L115-116: all the roles of melanins and neuromelanins could be cited: the binding of heavy metal is detrimental in long term but be a safe way of discard heavy metals in cells.

See also: (Ito, Wakamatsu et al. 2018)

L138: a reference is missing for the sentence: “It seems that neuromelanin is a by-product of dopamine 138 metabolism formed in the autooxidation pathway, without the key role of UV radiation 139 in this process (opposite to skin melanin), due to the obvious limitation of UV access to 140 neurons in midbrain.”

L153: a reference is missing: Neuromelanin concentration fluctuates in the aging brain. It seems that exceeding a 153 certain concentration threshold of this dye in dopaminergic neurones may trigger the neu-154 rodegeneration.

L164: this sentence is not clear: “However, the binding of toxins within neuro-melanin causes their concentration may reach higher values precisely in neurones rich in this pigment.”

L207: this sentence is not clear, because not all cancers are related to melanin: “This result seems logical, because cancer is a proliferative disease of melanin-containing cells caused by excessive cell division, while PD, as a degenerative disease (also containing melanin), is dominated by cell loss.”

L268: is it 21th century? And already said.

L273: idem

 Ito, S. and K. Wakamatsu (2008). "Chemistry of mixed melanogenesis--pivotal roles of dopaquinone." Photochem Photobiol 84(3): 582-592.

Ito, S., K. Wakamatsu and T. Sarna (2018). "Photodegradation of Eumelanin and Pheomelanin and Its Pathophysiological Implications." Photochem Photobiol 94(3): 409-420.

Comments on the Quality of English Language

The English should be checked some sentences were not understood.

Author Response

Major comments

The authors reviewed the literature on melanin and neuromelanin. The melanin pathway is not well described and need a better description.

The melanin synthesis pathway has been the main topic of several research and review papers. The phrase "melanin" and "synthesis" in the title and abstract leads to 3,222 publications in the PubMed database alone. We did not want to duplicate the information present in the literature, so we decided to present a schematic presentation of this process, without structural formulas, pointing only to the key elements of the pathway. Additionally, we believe that the proposed simple schemes have a high potential for citation, as many other publications describe the synthesis of melanin in great detail, using structural formulas.

There is no method section with a clear description of which years were considered with which criteria, instead of writing them in the text. The medical report of the literature on the effect of PD treatment on the melanin pathways and melanoma should be more rigourous, with exactly for each treatment, how many paper were considered, how many reports with positive and negative results, this could be reported in a table with the number of patients, the reference included. Then a discussion of the results can be included.

Thank you very much for your comment. Our work is not a typical systematic review because the topic presented is extremely heterogeneous. Furthermore, you can only find single articles related to the treatment of PD and melanoma, in which research was conducted at various levels (in vitro, in vivo, clinical observations). These studies cannot be compared with each other to prepare meta-analysis. That is why we decided to do a narrative review. Among the drugs used in PD, we selected those that have a direct impact on dopamine concentration by increasing it. Other drugs, eg, dopaminergic receptor agonists, were not considered.

Minor comments

L41: the authors are general but talk about neuromelanin.

The initial stages of skin melanin and neuromelanin synthesis are the same, although they can be catalyzed by other enzymes (e.g. EC 1.14.18.1 - tyrosinase (EC 1.14.18.1) or tyrosine 3-monooxygenase). Sentence 41 in the introduction is general and may refer to both relationships.

L47: « just two types of melanin” it is not totally correct because eumelanin is a mixture of nitrogenous polymers of DHI and DHICA, for reference see e.g.: ( Ito and Wakamatsu 2008). Melanin pathway could be better explained.

Thank you for your attention. We agree that the given sentence is too general. It has been corrected according to the reviewer's suggestion. Proper reference was added.

L47: the sentence on the PD symptoms could start a new paragraph.

The sentence was corrected

L101: Pheomelanin requires sulfhydryl group of any origin, e.g.: glutathione or cysteine.

The sentence was corrected.

L108: reference is missing.

The references are identical to the references of the next sentence. Both sentences describe the same experiment by Lembo et al.

 L108: melanin is not a protein.

 That's true. Our oversight. The sentence has been corrected.

L111: the authors could be cited instead of the publication journal.

The sentence was corrected.

L115-116: all the roles of melanins and neuromelanins could be cited: the binding of heavy metal is detrimental in long term but be a safe way of discard heavy metals in cells.

See also: (Ito, Wakamatsu et al. 2018)

The sentence was corrected and a proposed reference was added.

 L138: a reference is missing for the sentence: “It seems that neuromelanin is a by-product of dopamine 138 metabolism formed in the autooxidation pathway, without the key role of UV radiation 139 in this process (opposite to skin melanin), due to the obvious limitation of UV access to 140 neurons in midbrain.”

The appropriate reference was added

L153: a reference is missing: Neuromelanin concentration fluctuates in the ageing brain. It seems that exceeding a 153 certain concentration threshold of this dye in dopaminergic neurones may trigger the neu-154 rodegeneration.

The reference corresponds to the first sentence (neuromelanin concentration fluctuates in the aging brain). The second sentence is our speculation. The reference was moved into the correct location.

L164: this sentence is not clear: “However, the binding of toxins within neuro-melanin causes their concentration may reach higher values precisely in neurones rich in this pigment.”

The sentence was reedited

L207: this sentence is not clear, because not all cancers are related to melanin: “This result seems logical, because cancer is a proliferative disease of melanin-containing cells caused by excessive cell division, while PD, as a degenerative disease (also containing melanin), is dominated by cell loss.”

The sentence was corrected to suggest that all cancer is related to the growth of melanin-producing cells.

L268: is it 21th century? And already said.

In this review, we focused on publications from the last 20 years. However, the sentence mentioned by the reviewer concerns the last 30 years of the previous century, in which there were also reports of a possible relationship between L-DOPA treatment and the occurrence of melanoma.

 L273: idem

As we mentioned above, our reviews covered the last 20 years. The sentence is correct.

Reviewer 3 Report

Comments and Suggestions for Authors

The Authors performed a study about the role of melanin (and its synthesis) in PD and the correlation between PD and melanoma, where above all PD treatments may play a pivotal role in the development of melanoma in PD under treatments. The study is of interest and well writte. Minor changes are needed:

- Page 2 lines 51-52: please remove the sentences with the questions marks. 

- Did you find in the literature also informations regarding an increased risk of dysplastic nevi in PD patients under treatment? In case, please add.

- Did you find some information about other cutaneous malignancies.

-  Can you speculate also about possible association between melanoma and other neurological diseases and their treatments? 

- I suggest to add some sentence also about Vitamin D synthesis. Indeed Vitamin D deficency plays a pivotal role in PD (as well in other neurodegenerative diseases), as reported by Behl T, et al. (Pharmacol. 2022 Dec 19;13:993033). At the same time deficency of Vitamin plays a pivotal role also in melanoma, with some difference accoridng to the anatomic site as reported in this article "Clinicopathological features, vitamin D serological levels and prognosis in cutaneous melanoma of shield-sites: an update. Med Oncol. 2015 Jan;32(1):451. doi: 10.1007/s12032-014-0451-4. Epub 2014 Dec 17. PMID: 25516505." Please add a paragraph about this aspect and add these references. 

Author Response

The study is of interest and well writte.

Thank you very much for your positive assessment of our work.

Minor changes are needed:

- Page 2 lines 51-52: please remove the sentences with the questions marks. 

The indicated text was corrected

- Did you find in the literature also informations regarding an increased risk of dysplastic nevi in PD patients under treatment? In case, please add.

Searching the PubMed database, there are no publications referring to "parkinson's" and "dysplastic nervosa". In Google Scholar you can find articles that refer to the keywords mentioned above, but they are not related to the treatment of PD with the drugs described in this study.

 - Did you find some information about other cutaneous malignancies.

Studies assessing the incidence of cancer in patients with PD show that the incidence is reduced. But this does not apply to all cancers. Melanoma, but also non-melanoma cancers, occur more often than in the general population. However, data for non-melanoma skin cancer are less reliable than data for melanoma. The causal factors for this remain unknown.

(e.g. Ferreira et al. Skin cancer and Parkinson's disease. Mov Disord. 2010 Jan 30;25(2):139-48)

-  Can you speculate also about possible association between melanoma and other neurological diseases and their treatments? 

This topic is very extensive and goes beyond the scope of this work. When searching for specific diseases in the PubMed database, the word "melanoma" cooccurs with "multiple sclerosis" (title / abstract) 157 times, with "dementia" 37 times, with "ischemic stroke" 28 times, and "epilepsy" 49 times. Therefore, we will consider preparing a separate paper on this topic.

- I suggest to add some sentence also about Vitamin D synthesis. Indeed Vitamin D deficency plays a pivotal role in PD (as well in other neurodegenerative diseases), as reported by Behl T, et al. (Pharmacol. 2022 Dec 19;13:993033). At the same time deficency of Vitamin plays a pivotal role also in melanoma, with some difference accoridng to the anatomic site as reported in this article "Clinicopathological features, vitamin D serological levels and prognosis in cutaneous melanoma of shield-sites: an update. Med Oncol. 2015 Jan;32(1):451. doi: 10.1007/s12032-014-0451-4. Epub 2014 Dec 17. PMID: 25516505." Please add a paragraph about this aspect and add these references. 

The role of vitamin D in metabolism is a very interesting and not fully understood issue. The role of this vitamin in PD and melanoma is also the subject of many studies. Since this is not the main topic of our work, we did not insert a new chapter, but we added information on this topic to one of the paragraphs. Together with the recommended literature.